

# Transport fluctuations in integrable models out of equilibrium

**Jason Myers[1], M. J. Bhaseen[2], Rosemary J. Harris[3], and Benjamin Doyon[1⋆]**

**1** Department of Mathematics, King's College London, Strand, London WC2R 2LS, U.K.
**2** Department of Physics, King's College London, Strand, London WC2R 2LS, U.K.
**3** School of Mathematical Sciences, Queen Mary University of London,
Mile End Road, London E1 4NS, U.K.

⋆ benjamin.doyon@kcl.ac.uk

## Abstract

We propose exact results for the full counting statistics, or the scaled cumulant generating function, pertaining to the transfer of arbitrary conserved quantities across an interface in homogeneous integrable models out of equilibrium. We do this by combining insights from generalised hydrodynamics with a theory of large deviations in ballistic transport. The results are applicable to a wide variety of physical systems, including the Lieb-Liniger gas and the Heisenberg chain. We confirm the predictions in non-equilibrium steady states obtained by the partitioning protocol, by comparing with Monte Carlo simulations of this protocol in the classical hard rod gas. We verify numerically that the exact results obey the correct non-equilibrium fluctuation relations with the appropriate initial conditions.



# 1 Introduction

Many-body physics far from equilibrium poses some of the most challenging questions in modern science [1]. It has attracted a large amount of attention in recent years with, for instance, experimental observations of quantum heat flow [2, 3] and investigations into the processes of thermalisation in isolated systems [4]. In one dimension, integrability strongly affects non-equilibrium physics, as demonstrated in the seminal quantum Newton's cradle experiment on cold atomic gases [5]. Relaxation to stationary states is constrained by the macroscopic number of conservation laws afforded by integrability [6–10]. As quantum transport problems are accessible via hydrodynamics [11–20], an emergent, large-wavelength hydrodynamic theory for integrable systems has been proposed, generalised hydrodynamics (GHD) [21–24]. It accounts for the macroscopic number of interacting ballistic currents. GHD has been directly tested in a neoteric experiment [25], and gives rise to a panoply of results which are expected to be exact, including non-equilibrium flows [21,22,26–29], Drude weights [30–33] and large-scale correlations [34,35], as well as a hydrodynamic-scale solution to the quantum Newton's cradle setup at arbitrary coupling strength [36].

A full characterisation of non-equilibrium states, however, must go beyond the study of relaxation processes and hydrodynamics, and one of the most important challenges is to provide organising principles with universal and widely applicable reach. In equilibrium, a powerful description is that centred on the analysis of fluctuations of thermodynamic quantities through statistical-mechanical ensembles and free energies. Out of equilibrium, the presence of non-zero currents suggests that a study of dynamical fluctuations might provide a similar level of understanding [37–41]. This line of thought has led to a large deviation framework for non-equilibrium statistical mechanics [42]. For instance, the so-called large-deviation function, which describes the rate of occurrence of rare but large fluctuations, plays the role of an entropy. The related scaled cumulant generating function (SCGF) for full counting statistics plays the role of a free energy.[1] It is of paramount importance to obtain exact results for such

---

[1] In keeping with the nomenclature used in the literature on quantum transport, in this work we use interchangeably the terminology "SCGF" and "full counting statistics" in the context of large-time, scaled transport fluctuations.

functions in transport setups of truly interacting many-body models in order to gain a deeper understanding of non-equilibrium physics.

Fluctuations in non-equilibrium transport can be studied by analysing the statistics of the number of particles, their energy, or any charge they carry, passing through an interface in a bipartition of the system, see Figure 1. Exact results for transport SCGFs have been obtained in various systems (at various levels of mathematical rigour). For instance, exact formulae are known in non-equilibrium steady states (NESSs) of some stochastic classical gases such as exclusion processes [37, 40, 41]; these are understood within macroscopic fluctuation theory [43–47] based on diffusive hydrodynamics. Some results have also been obtained in open quantum chains, see e.g. [48, 49]. Such stochastic or open models, however, make assump-

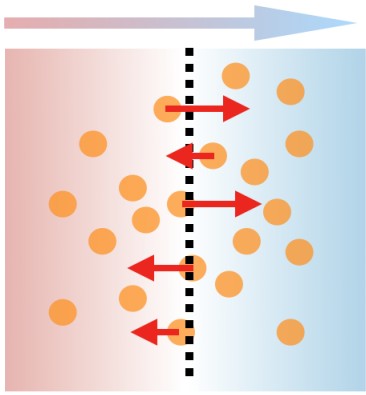

Figure 1: Schematic illustration depicting counting statistics in a steady state regime. One "counts" the number of particles (or their energy or charge) passing a given coordinate during a large time interval and then gathers the statistics of these large numbers, scaled with time.

tions about the external baths. It is crucial to understand intrinsic transport fluctuations in *deterministic,* isolated, quantum and classical systems, where exact many-body interactions are fully taken into account. In an ensemble formulation, fluctuations originate from those in the initial state. Despite many efforts, only a few results exist: free-fermions with the celebrated Lesovik-Levitov formula [50–52], harmonic chains [53] and free field theory [54, 55], particular integrable impurity models [56], and one-dimensional critical systems [57, 58]; see the review [59]. Some results also exist for fluctuation statistics of other quantities, not related to transport, in certain integrable models, see for instance [60–63]. A full grasp of counting statistics for transport in interacting many-body systems, especially where integrability and ballistic processes dominate, remains an open problem.

In this work, we obtain the first (to our knowledge) model-agnostic exact expression for transport SCGFs in homogeneous stationary states of interacting one-dimensional integrable systems. Our analytical approach involves a new and completely general framework based on large deviation theory and Euler-scale linear fluctuating hydrodynamics that gives access to exact SCGFs for ballistic transport, developed in a companion paper by two of us (BD and JM) [92].

The states considered are very general, and include current-carrying NESSs. In particular, they include NESSs obtained by the partitioning protocol [15, 16, 19–22, 57–59, 64, 65, 67–71], where an inhomogeneous, "unbalanced" initial condition generates, at large times, a homogeneous stationary current-carrying state. We emphasise that although the theory applies to transport statistics in homogeneous stationary states, this statistics can be evaluated from transfer measurements starting at the initial time of the protocol, as the large-time, stationary region dominates. The expression applies to all models whose large-scale dynamics is gov-

erned by GHD, and to transport of all local conserved quantities they admit. This includes the Lieb-Liniger model [72,73] which has been shown to describe cold atomic gases in quasi-one-dimensional traps [74–76] (see e.g. the experiments [5, 25] where the integrability of the Lieb-Liniger model played an important role, and the book [77] for a review), and many other quantum field theories, as well as integrable quantum chains, classical field theories, and classical gases such as the hard rod gas [78,79] and soliton gases [80–84].

We provide explicit checks on the predictions for the first few transport cumulants by comparing with Monte Carlo simulations of the hard rod gas. For this purpose, we explicitly implement the partitioning protocol in the hard rod gas, and measure the total energy transferred from the initial time of the protocol. We find very convincing agreement with the theory. We also verify numerically that the exact expression in the Lieb-Liniger model satisfies the non-equilibrium fluctuation relations of Gallavotti-Cohen type [85–90].

The paper is organised as follows. In section 2 we review the large deviation theory for non-equilibrium transport. In section 3 we outline the general theory of fluctuations in ballistic transport based on Euler-scale identities. In section 4 we review aspects of the thermodynamic Bethe ansatz that will be useful in this work. In section 5 we present our main result, the exact full counting statistics for transport in integrable models. In section 6 we apply this result to the hard rod gas, providing Monte Carlo verification, and in section 7 we apply it to the Lieb-Liniger model, verifying the non-equilibrium fluctuation relations. Finally, we conclude in section 8. A series of appendices provide supporting calculations.

## 2 Large deviation theory for transport

This section provides background on large deviation theory (LDT), and introduces the scaled cumulant generating function (SCGF) for transport of conserved quantities in the context of integrable models. We also briefly recall some aspects of non-equilibrium fluctuation relations (FRs). The setup is as in fig. 1, specialised to one dimension – the interface is then a single point.

### 2.1 Rate functions and multiple conservation laws

Large deviation theory focuses on fluctuating quantities $J^{(t)}$ which are extensive with respect to some parameter $t$, and whose densities $J^{(t)}/t$ take almost-sure values $\bar{j}$ in the extensive limit $t \to \infty$. A standard example is the energy in equilibrium thermodynamics, with $t$ being the volume. According to the large deviation principle [42], such extensive quantities have probability distributions that are exponentially peaked at the almost-sure value; in the cases of interest here, this takes the form

$$P(J^{(t)} = tj) \sim e^{-tI(j)}, \quad \text{where} \quad \begin{cases} I(j) = 0 & j = \bar{j} \\ I(j) > 0 & j \neq \bar{j} \end{cases}. \tag{1}$$

The function $I(j)$ is referred to as the large-deviation rate function. It describes the probabilities of rare but significant events where the quantity $J^{(t)}$ deviates "macroscopically" from $t\bar{j}$.

The framework is general enough to encompass fluctuations in transport, and systems that are far from equilibrium, see e.g. [91] for stochastic processes. In the setup of fig. 1 which we consider in the present paper, $J^{(t)}$ is the total current of some conserved charge that has passed through the interface in a time $t$, and the state is a homogeneous steady state. Recall that the evolution is deterministic, but the state, at time $t$, is fluctuating due to the ensemble

description of the initial condition leading to the probability distribution we denote as $P(\cdots)$ above.

In order to describe more precisely the state and the total current $J^{(t)}$, we consider a system with a certain number of (local or quasi-local) conserved quantities $Q_i = \int \mathrm{d}x\, \mathsf{q}_i(x,0)$. This number will be taken to infinity, as we are interested in integrable models which, in the thermodynamic limit, admit an infinite number of conserved quantities. Consider the associated local conservation laws

$$\partial_t \mathsf{q}_i(x,t) + \partial_x \mathsf{j}_i(x,t) = 0, \tag{2}$$

indexed by $i$, with current density $\mathsf{j}_i$. States where entropy is maximised with respect to all local conservation laws are characterised by as many Lagrange parameters $\beta^i$ as there are conservation laws and, formally, have probability measure or density matrix proportional to

$$e^{-\sum_i \beta^i Q_i}. \tag{3}$$

In integrable systems, these probability measures are referred to as generalised Gibbs ensembles (GGEs) [6–10]. The infinite sum over the conserved quantities in (3) must be dealt with carefully: a precise definition of GGEs requires an appropriate completion of the space of conserved quantities [9]. In this sense, $\sum_i \beta^i Q_i$ should be understood as a particular "pseudolocal" charge, characterised by coefficients $\beta^i$ in some basis decomposition. As is customary, we refer to GGEs with this general understanding. We will denote averages by $\langle\cdots\rangle_{\underline{\beta}}$ where $\underline{\beta}$ is the vector of Lagrange parameters $\beta^i$. These are the homogeneous steady states that we will concentrate on in this paper.

GGE states include many (homogeneous) NESSs. Indeed, the fundamental characteristic of a non-equilibrium steady state is that, despite being stationary, it breaks time-reversal invariance. As integrable systems admit conserved charges that break time-reversal symmetry, including the total momentum, many GGEs are non-equilibrium, current-carrying states. Physically, this corresponds to the fact that the presence of appropriate conserved quantities allows ballistic propagation, where currents are sustained without the need for an external force. A paradigmatic example of a NESS is that emerging from the partitioning protocol [64, 65], which is referred to as the Riemann problem in hydrodynamics (see e.g. the lecture notes [66]). In this protocol, the steady state is formed, at very large times, by deterministic or unitary evolution from an initial inhomogeneous state which is homogeneous far to the left and right ($x \to \pm\infty$), as $e^{-\sum_i \beta_l^i Q_i}$ (left) and $e^{-\sum_i \beta_r^i Q_i}$ (right). Despite the inhomogeneous initial condition, at infinite times, in any finite region around the central position, the state is expected to become homogeneous. In integrable systems, this state is a GGE. Such NESSs emerging from the partitioning protocol were constructed for integrable models in [21, 22, 26]: the $\beta^i$'s for the NESS were obtained as functions of the $\beta_l^i$'s and $\beta_r^i$'s of the initial condition of the protocol. We will make use of these results below.

We focus on the total transfer of some particular charge $Q_{i_*}$, say from the left to the right of the system, in time $t$, see fig. 1. This can be, for instance, the number of particles (if particle number is conserved), the electric charge (in systems with $U(1)$ symmetry), the energy, or any other conserved quantity. We are then interested in the total current passing by the origin,

$$J^{(t)} = \int_0^t \mathrm{d}s\, \mathsf{j}_{i_*}(0,s). \tag{4}$$

One expects (1) to hold for this quantity.

## 2.2 Scaled cumulant generating function and fluctuation relations

In a NESS, the average of $J^{(t)}/t$ is given by the almost-sure value $\bar{j}$. More generally, consider the scaled cumulants $c_n$ of the transferred quantity, or rather their generating function, the SCGF:

$$F(\lambda) = \lim_{t\to\infty} \frac{1}{t} \log \langle e^{\lambda J^{(t)}} \rangle_{\underline{\beta}} = \sum_{k=1}^{\infty} \frac{\lambda^k}{k!} c_k. \tag{5}$$

By the Gärtner-Ellis theorem, if this limit exists and the result is differentiable, then the Legendre-Fenchel transform of $F(\lambda)$ gives the large deviation rate function $I(j)$ (see e.g. [42]). Note that if in fact the limit exists at each order in $\lambda$, then all the cumulants of $J^{(t)}$ scale like $t$, that is

$$c_k = \lim_{t\to\infty} \frac{1}{t} \langle [J^{(t)}]^k \rangle^{\mathrm{c}}_{\underline{\beta}}, \tag{6}$$

where the superscript $c$ means that these are connected averages. The scaled cumulants can be expressed in terms of the average current and its connected time-integrated correlation functions. For example[2]

$$c_1 = \langle j_{i_*} \rangle_{\underline{\beta}}, \quad c_2 = \int dt \, \langle j_{i_*}(0,t) j_{i_*}(0,0) \rangle^{\mathrm{c}}_{\underline{\beta}}, \quad c_3 = \int dt dt' \, \langle j_{i_*}(0,t') j_{i_*}(0,t) j_{i_*}(0,0) \rangle^{\mathrm{c}}_{\underline{\beta}}, \tag{7}$$

where we recall that $i_*$ is the index of the charge, and corresponding current, we are interested in. The quantity $c_2$ is referred to as the zero-frequency noise in mesoscopic physics, and was dubbed the Drude self-weight in [32]. In sections 6 and 7 we will consider energy transfer, making $j_{i_*}$ more concrete.

Key ingredients of the general theory of NESSs are fluctuation relations, which compare the probabilities of "forward" and "backward" currents; see the reviews for classical [38, 93–95] and quantum [39, 96, 97] systems. For currents obeying (1), the fluctuation relations are reflected in fundamental symmetries of the SCGF connecting scaled cumulants in a non-trivial way:

$$F(\lambda) = F(\nu - \lambda), \tag{8}$$

where $\nu$ is a constant encoding properties of the force or external baths generating the NESS. This formula applies, for instance, in the NESS emerging from the partitioning protocol described in subsection 2.1. Indeed, under certain conditions – if both the dynamics and the charge $Q_{i_*}$ are time-reversal invariant and the initial state has an imbalance in $Q$ only, $\beta_l^i = \beta_r^i \; \forall \, i \neq i_*$ – then we expect (8) to hold with $\nu = \beta_l^{i_*} - \beta_r^{i_*}$ [98, 99].

## 3 Fluctuations from Euler-scale hydrodynamics

The general theory of fluctuations for one-dimensional systems supporting ballistic transport, which we will refer to as the ballistic fluctuation formalism, is developed in [92]. This is the approach that we will use in order to access fluctuations in integrable systems. Explicitly, the theory shows how to construct $F(\lambda)$, as defined in subsection 2.2, using Euler-scale identities. Possible corrections to the Euler scale of diffusive and other types would provide subleading corrections to ballistic fluctuations

---

[2]In field theory, UV divergences occur at coincident points. However, the scaled cumulants are independent of such divergences and thus UV finite [59], as can be seen by using the conservation laws (2) to change the positions of the currents to different space coordinates.

## 3.1 Flux Jacobian

Consider the averages of all local conserved densities, $\langle \mathfrak{q}_i \rangle = \langle \mathfrak{q}_i \rangle_{\underline{\beta}} = \langle \mathfrak{q}_i(0,0) \rangle_{\underline{\beta}}$, and the current averages $\langle \mathfrak{j}_i \rangle = \langle \mathfrak{j}_i \rangle_{\underline{\beta}} = \langle \mathfrak{j}_i(0,0) \rangle_{\underline{\beta}}$. We can write $\langle \mathfrak{j}_i \rangle$ as functions of $\langle \mathfrak{q}_i \rangle$ by inverting the relation between $\langle \mathfrak{q}_i \rangle$ and the Lagrange parameters $\beta^i$. This functions describe the equations of state: in this form currents are sometimes referred to as "fluxes". From the equations of state we construct the flux Jacobian

$$A_i^{\ j} = \frac{\partial \langle \mathfrak{j}_i \rangle}{\partial \langle \mathfrak{q}_j \rangle}. \tag{9}$$

As this matrix plays a fundamental role in the ballistic fluctuation formalism, it is useful to recall some of the important equations where it is involved. The physical interpretation of the flux Jacobian is that it describes the flow of conserved quantities within maximal entropy states. Indeed, first, it is naturally involved in the Euler hydrodynamic equations for the system. These are the equations for the evolution of large-wavelength, low-frequency inhomogeneous states, and are obtained by applying the conservation laws (2) on densities and currents averaged within local entropy-maximised states [100]. Using the flux Jacobian, the Euler equations are written as equations for the averages $\langle \mathfrak{q}_i \rangle_{\underline{\beta}(x,t)}$,

$$\partial_t \langle \mathfrak{q}_i \rangle_{\underline{\beta}(x,t)} + \sum_j A_i^{\ j} \partial_x \langle \mathfrak{q}_j \rangle_{\underline{\beta}(x,t)} = 0. \tag{10}$$

By linear-response theory, $A_i^{\ j}$ is also involved in evolution equations for space-time dependent connected correlation functions within stationary, homogeneous, maximal entropy states [100],

$$\partial_t \langle \mathfrak{q}_i(x,t) \mathfrak{q}_k(0,0) \rangle_{\underline{\beta}}^{\mathrm{c}} + \sum_j A_i^{\ j} \partial_x \langle \mathfrak{q}_j(x,t) \mathfrak{q}_k(0,0) \rangle_{\underline{\beta}}^{\mathrm{c}} = 0. \tag{11}$$

Correspondingly, it is related to the rate at which correlation functions spread [101],

$$\lim_{t \to \infty} \int \mathrm{d}x \, \frac{x}{t} \langle \mathfrak{q}_i(x,t) \mathfrak{q}_k(0,0) \rangle_{\underline{\beta}}^{\mathrm{c}} = \sum_j A_i^{\ j} \int \mathrm{d}x \, \langle \mathfrak{q}_j(x,t) \mathfrak{q}_k(0,0) \rangle_{\underline{\beta}}^{\mathrm{c}}. \tag{12}$$

The equations above lead to the observation that eigenvalues of the flux Jacobian correspond to velocities at which perturbations travel within a homogeneous state.

## 3.2 Flow equation and SCGF

Two equations form the backbone of the ballistic fluctuation formalism, the flow equation and an equation showing how this can be used to obtain the SCGF. We start with the flow equation. In the ballistic fluctuation formalism, one defines the state $\langle \mathcal{O} \rangle (\lambda)$ by biasing the measure $e^{-\sum_i \beta^i Q_i}$, multiplying it by the operator generating the charge transport of interest, $e^{\lambda \int \mathrm{d}t \, j_{i_*}(0,t)}$. Here $\lambda$ should be seen as the conjugate parameter for the particular quantity of interest indexed by $i_*$ as per (5) with (4).[3] It turns out that a state defined in this way is still a stationary, homogeneous, maximal entropy state [92]. The flux Jacobian is used in order to define a relation that characterises how the conserved quantities are affected by this $\lambda$-modification, a flow equation. This equation takes the form [92]

$$\partial_\lambda \langle \mathfrak{q}_j \rangle (\lambda) = \sum_i \mathrm{sgn}(A(\lambda))_{i_*}^{\ i} \int \mathrm{d}x \, \langle \mathfrak{q}_j(x,0) \mathfrak{q}_i(0,0) \rangle^{\mathrm{c}}(\lambda), \tag{13}$$

---

[3]Strictly speaking, one should use $\lambda^{i_*}$ as a variable specifically linked to $j_{i_*}$ but this creates tortuous notation.

where sgn($A$) is the matrix obtained by diagonalising $A$ and taking the sign of its eigenvalues.

A motivation for this result can be obtained as follows. For simplicity let us consider the case where a single conserved quantity is present, so that $A$ is a number, the velocity of a perturbation within the homogeneous state. By definition of the $\lambda$-modified state, $\partial_\lambda \langle j \rangle(\lambda) = \int dt \, \langle j(0,t)j(0,0) \rangle^c(\lambda)$. On the right-hand side, by linear response in the limit $x, t \to \infty$ with $x/t$ constant (Euler scaling limit), the current is related to the corresponding charge multiplied by its velocity through the medium, so we can replace $j(x,t)$ by $Aq(x,t)$. Solving the hydrodynamic equation (11) by the method of characteristics, the correlation function is supported on $x = At$, hence we can use $dx = |A|dt$. Therefore, $\int dt \, \langle j(0,t)j(0,0) \rangle^c(\lambda) = A^2/|A| \int dx \, \langle q(x,0)q(0,0) \rangle^c(\lambda)$. On the other hand, $\partial_\lambda \langle j \rangle(\lambda) = A \partial_\lambda \langle q \rangle(\lambda)$. This gives (13) specialised to the case of a single conserved quantity.

It is useful to recast (13) in a different form so as to expose the $\lambda$-dependence in the Lagrange multipliers:

$$\partial_\lambda \beta^i(\lambda) = -\text{sgn}(A(\lambda))^i_{i_*}. \tag{14}$$

This is the flow equation at the basis of the general theory.

The flow equation can be used to determine the SCGF. This is achieved by first solving for $\beta^i(\lambda)$ and considering the $\lambda$-dependent currents $\langle j_i \rangle_{\underline{\beta}(\lambda)} \equiv \langle j_i \rangle(\lambda)$. Then, the general theory predicts that [92],

$$F(\lambda) = \int_0^\lambda d\lambda' \, \langle j_{i_*} \rangle(\lambda'). \tag{15}$$

This equation generalises the relationship between $F(\lambda)$ and the cumulants (see (5) and (7)) for $\lambda \neq 0$.

The average currents $\langle j_i \rangle$ and the flux Jacobian $A_i^j$ are known exactly in integrable models, see [21, 22, 102] and [32]. Combining these exact results with the above formalism[4] gives an expression for $F(\lambda)$. First, we must review the Thermodynamic Bethe Ansatz (TBA) formalism. The TBA description of integrable models will then be amenable to the application of the ballistic fluctuation formalism.

## 4 Thermodynamic Bethe Ansatz

In this section we review the powerful description of integrable models in the thermodynamic limit using the language of the Thermodynamic Bethe Ansatz (TBA).

In a wide family of many-body integrable systems, GGEs (introduced in section 2) are efficiently described via the TBA [103, 104] in terms of "quasiparticles", whose scattering is elastic and factorises into two-body processes. The TBA is powerful enough to describe thermodynamics of not only quantum, but also classical models (including field theories, gases and chains). Exact solutions for the NESSs from the partitioning protocol in integrable models were expressed in the language of TBA in [21, 22, 26]. Quasiparticles are parametrised by a spectral parameter $\theta$, which, in general, encodes both their momenta and type. Here, for simplicity, we will consider $\theta \in \mathbb{R}$ with a single particle type. Each quasiparticle $\theta$ carries a quantity $h_i(\theta)$ of charge $Q_i$, for instance momentum and energy, which we will denote by $p(\theta)$

---

[4]The validity of the ballistic fluctuation formalism rests on the assumption of strong enough decay of current correlation functions at large times [92]: essentially the time-integrated correlation functions (7) should be convergent. In integrable models, dynamical two-point correlation functions generically decay, at large times, as $1/t$ [32, 34]. However, from [32, Eq 4.55], one can see that current two-point functions *decay more rapidly along the time direction*, making them integrable, with finite value given by [32, Eq 1.4]. Likewise, higher-point correlation functions are expected to decay strongly enough [34]. The arguments for the ballistic fluctuation formalism in fact require, in their current form, strong enough decay in a neighbourhood of the time direction [92]; a precise analysis of this in integrable models is left for future works.

and $E(\theta)$ respectively. The generalised specific free energy is given by $\int \frac{\mathrm{d}p(\theta)}{2\pi} \mathsf{F}(\epsilon(\theta))$ where the pseudoenergy $\epsilon(\theta)$ involves the Lagrange parameters and satisfies the non-linear integral equation

$$\epsilon(\theta) = \sum_i \beta^i h_i(\theta) + \int \frac{\mathrm{d}\alpha}{2\pi} \varphi(\theta, \alpha) \mathsf{F}(\epsilon(\alpha)). \tag{16}$$

Here, the "differential scattering phase" $\varphi(\theta, \alpha)$ encodes the microscopic interactions, and is related to the two-body scattering matrix. We assume $\varphi(\theta, \alpha)$ to be symmetric for simplicity – in many integrable models, there is a choice of spectral parameter where this is the case. The TBA free energy function $\mathsf{F}(\epsilon)$ depends on the statistics of the quasiparticles: for instance, $-\log(1+e^{-\epsilon})$ for fermions, $\log(1-e^{-\epsilon})$ for bosons, $-e^{-\epsilon}$ for classical particles, $1/\epsilon$ for classical radiative modes [34]. Intuitively, the pseudoenergy $\epsilon(\theta)$ of a quasiparticle $\theta$ determines the amount this quasiparticle contributes to the GGE probability distribution. It is a modification of the form it would have in free theories that takes into account the interaction with the other quasiparticles. It is important to note that instead of the Lagrange parameters $\beta^i$, the pseudoenergy can also be used to fully characterise the GGE.

The equations of state in GGEs were found in [21, 22] and proven for relativistic field theory in [102], with the exact currents written as

$$\langle \mathsf{j}_i \rangle = \int \frac{\mathrm{d}\theta}{2\pi} E'(\theta) n(\theta) h_i^{\mathrm{dr}}(\theta), \quad n(\theta) = \frac{\mathrm{d}\mathsf{F}(\epsilon)}{\mathrm{d}\epsilon}\bigg|_{\epsilon=\epsilon(\theta)}, \quad E'(\theta) = \frac{\mathrm{d}E(\theta)}{\mathrm{d}\theta}. \tag{17}$$

Here $n(\theta)$ is referred to as the occupation function, and, like the pseudoenergy, is also sufficient to fully characterise the GGE. The superscript "dr" in $h_i^{\mathrm{dr}}$ refers to a fundamental operation within the TBA formalism, the *dressing transformation*. In a similar fashion to the pseudoenergy, the dressing transformation modifies a bare quantity, in this case $h_i$, by taking into account the interaction with the other quasiparticles. The dressing transformation of some quantity $g(\theta)$ is defined through the linear integral equation

$$g^{\mathrm{dr}}(\theta) = g(\theta) + \int \frac{\mathrm{d}\alpha}{2\pi} \varphi(\theta, \alpha) n(\alpha) g^{\mathrm{dr}}(\alpha). \tag{18}$$

Using the dressing transformation, the flux Jacobian $A_i{}^j$, introduced in the previous section, was evaluated exactly in [32]. A consequence of the expression found there is that $A_i{}^j$ is diagonalised by the dressing transformation: the vectors $h_i^{\mathrm{dr}}(\theta)$, for fixed $\theta$, are its eigenvectors. The corresponding eigenvalues are the *effective velocities* of the generalised fluids, given by $v^{\mathrm{eff}}(\theta) = (E')^{\mathrm{dr}}/(p')^{\mathrm{dr}}$. Effective velocities describe the velocity of quasiparticles through the medium, given the interactions with other quasiparticles. More precisely, the result of [32] can be recast into the explicit form

$$A_i{}^j = \int \mathrm{d}\theta \, h_i^{\mathrm{dr}}(\theta) v^{\mathrm{eff}}(\theta) h_{\mathrm{dr}}^j(\theta), \tag{19}$$

where here $h_{\mathrm{dr}}^j(\theta)$ are the orthonormal conjugates to $h_i^{\mathrm{dr}}$ under the $L^2(\mathbb{R})$ inner product, i.e., $\int \mathrm{d}\theta \, h_i^{\mathrm{dr}}(\theta) h_{\mathrm{dr}}^j(\theta) = \delta_i^j$ (with Kronecker delta, as the set of indices $i$ is taken to be discrete).[5] By assumed completeness of the set of functions $h_j^{\mathrm{dr}}(\theta)$, we have

$$\sum_j A_i{}^j h_j^{\mathrm{dr}}(\theta) = v^{\mathrm{eff}}(\theta) h_i^{\mathrm{dr}}(\theta). \tag{20}$$

---

[5] If $h^j(\theta)$ are defined to be orthonormal to $h_i(\theta)$, then these orthonormality relations define the "lower-index dressing" $h \mapsto h_{\mathrm{dr}}$, making it a different transformation from the usual "upper-index" dressing.

In the Euler fluctuation theory, we are interested in $\mathrm{sgn}(A)_i^{\ j}$, which is given by

$$\mathrm{sgn}(A)_i^{\ j} = \int \mathrm{d}\theta \, h_i^{\mathrm{dr}}(\theta) \, \mathrm{sgn}(v^{\mathrm{eff}}(\theta)) h_{\mathrm{dr}}^j(\theta). \tag{21}$$

The example systems used in this work are the classical hard rod gas in section 6, and the quantum Lieb-Liniger gas in section 7. In both cases, there is only a single quasiparticle type. Furthermore these are both Galilean systems, so that $\theta$ can be taken as the quasiparticle velocity. The quasiparticle momentum and energy are given by $p(\theta) = \theta$ and $E(\theta) = \theta^2/2$ respectively, where we set the mass to unity.

## 5 Exact transport fluctuations in integrable models

We now turn to writing (14) and (15) for integrable models. This gives us *an exact expression for the full counting statistics $F(\lambda)$ of any total current*, in terms of TBA quantities and a pseudoenergy function satisfying a flow-equation. From this, *all cumulants can be obtained order by order* in terms of TBA quantities in the original state. We believe this to be a remarkable result as, to our knowledge, no such widely applicable expression exists for interacting integrable models. The result naturally generalises the known expressions for free-fermion and other quadratic models [50–55], and for one-dimensional critical systems [57–59], see Appendix A and the discussion in [92]. However, its connection to the exact SCGF found in integrable impurity models [56] is not understood yet.

The exact expression for $F(\lambda)$ is presented in (25).

### 5.1 Full counting statistics

As per subsection 3.2, we consider $\beta$ becoming $\lambda$-dependent by satisfying the flow equation (14). Through (16) the pseudoenergy acquires a $\lambda$-dependence, $\epsilon(\theta; \lambda)$, and similarly all dressed functions become, $h^{\mathrm{dr}}(\theta; \lambda)$ through $n(\theta)$ in (17). From (16), the pseudoenergy $\epsilon(\theta; \lambda)$ satisfies

$$\epsilon(\theta; \lambda) = \sum_i \beta^i(\lambda) h_i(\theta) + \int \frac{\mathrm{d}\alpha}{2\pi} \varphi(\theta, \alpha) \mathsf{F}(\epsilon(\alpha; \lambda)). \tag{22}$$

Using Eq. (14), we find

$$\partial_\lambda \epsilon(\theta; \lambda) = -\sum_i \mathrm{sgn}(A(\lambda))_{i_*}^{\ i} h_i(\theta) + \int \frac{\mathrm{d}\alpha}{2\pi} \varphi(\theta, \alpha) n(\alpha; \lambda) \partial_\lambda \epsilon(\alpha; \lambda), \tag{23}$$

and therefore using the definition of the dressing operation and the eigenvalue equation (20),

$$\partial_\lambda \epsilon(\theta; \lambda) = -\sum_i \mathrm{sgn}(A(\lambda))_{i_*}^{\ i} h_i^{\mathrm{dr}}(\theta; \lambda) = -\mathrm{sgn}(v^{\mathrm{eff}}(\theta; \lambda)) h_{i_*}^{\mathrm{dr}}(\theta; \lambda). \tag{24}$$

As the pseudoenergy fully characterises the GGE, this defines the flow equation for integrable models.

An expression for $F(\lambda)$ can now be obtained using (15) and the solution of (24) in (17). We will show below that

$$
\begin{aligned}
F(\lambda) = -\int \frac{\mathrm{d}\theta}{2\pi} E'(\theta) \Big( &\mathrm{sgn}\big(v^{\mathrm{eff}}(\theta; \lambda)\big)\big(\mathsf{F}(\epsilon(\theta; \lambda)) - \mathsf{F}(\epsilon(\theta; 0))\big) \\
&- \sum_{\sigma \in \{\pm\}} \sum_{\tilde{\lambda} \in \lambda_*^\sigma(\theta) \cap [0,\lambda]} \sigma\big(\mathsf{F}(\epsilon(\theta; \tilde{\lambda})) - \mathsf{F}(\epsilon(\theta; 0))\big) \Big),
\end{aligned}
\tag{25}
$$

where the sets $\lambda_\star^\pm(\theta)$ are the turning points of the sign of the effective velocity:

$$\lambda_\star^\pm(\theta) = \{\lambda : v^{\text{eff}}(\theta;\lambda) = 0, \partial_\lambda v^{\text{eff}}(\theta;\lambda) \gtrless 0\}. \tag{26}$$

The proof of (25) proceeds as follows. The idea is to take a derivative of (25) with respect to $\lambda$ and show, in accordance with (15), equality with $\langle j_i \rangle$ from (17). We first take the derivative on the first line of the right-hand side of (25). Using the Leibniz Rule, two terms emerge. In the first, the derivative is applied on $\text{sgn}(v^{\text{eff}}(\theta;\lambda))$, giving a $\delta$-function contribution (under the integral), and thus the term

$$-\int \frac{\mathrm{d}\theta}{2\pi} E'(\theta) \delta(v^{\text{eff}}(\theta;\lambda)) \, \partial_\lambda v^{\text{eff}}(\theta;\lambda) \big( \mathsf{F}(\epsilon(\theta;\lambda)) - \mathsf{F}(\epsilon(\theta;0)) \big). \tag{27}$$

On the other hand, taking the $\lambda$-derivative on the second line of the right-hand side of (25), we also obtain a $\delta$-function contribution, which occurs when an element of the set $\lambda_\star^a$ enters the interval $[0,\lambda]$. This contribution therefore has the form

$$\int \frac{\mathrm{d}\theta}{2\pi} E'(\theta) \sum_{a\in\{\pm\}} \sum_{\tilde{\lambda}\in\lambda_\star^a(\theta)} \delta(\lambda - \tilde{\lambda}) a \big( \mathsf{F}(\epsilon(\theta;\tilde{\lambda})) - \mathsf{F}(\epsilon(\theta;0)) \big). \tag{28}$$

By a change of variables, we have

$$\delta(v^{\text{eff}}(\theta;\lambda)) = \sum_{a\in\{\pm\}} \sum_{\tilde{\lambda}\in\lambda_\star^a(\theta)} \frac{\delta(\lambda-\tilde{\lambda})}{|\partial_\lambda v^{\text{eff}}(\theta;\lambda)|} = \sum_{a\in\{\pm\}} \sum_{\tilde{\lambda}\in\lambda_\star^a(\theta)} \frac{a\delta(\lambda-\tilde{\lambda})}{\partial_\lambda v^{\text{eff}}(\theta;\lambda)} \tag{29}$$

and we see that (27) cancels (28). Finally, we are left with the second term from application of the Leibniz rule on the first line of the right-hand side of (25), the term in which the $\lambda$-derivative acts on the factor $\big( \mathsf{F}(\epsilon(\theta;\lambda)) - \mathsf{F}(\epsilon(\theta;0)) \big)$. Using (24), we obtain (17), with the $\lambda$-dependent state $n(\theta;\lambda)$. The final step is to check that $F(0) = 0$, which is trivially true. This completes the proof.

Eq. (25) is an exact general result for the SCGF – or full counting statistics – in GGEs of arbitrary integrable models. The key development, the inclusion of interactions, is contained within $v^{\text{eff}}(\theta;\lambda)$ and $\epsilon(\theta;\lambda)$. The form of the result separates the effects of the fluctuations in the state, encoded within the free energy function $\mathsf{F}(\epsilon)$, from the effects of the interactions. The state fluctuations give rise to transport fluctuations, but in a way that is affected by the interactions, as the quasiparticle velocities and charges depend on the fluctuating state. Explicitly, the function $F(\lambda)$ is obtained by solving (24) (numerically, or order by order in $\lambda$), and by then using the resulting pseudoenergy in order to evaluate the TBA quantities involved in (25), and integrating over the spectral parameter.

This result can be applied to any model with known TBA, opening up a diverse range of interacting quantum models for which we can obtain information about the statistics of flows. Importantly, the result agrees with the Lesovik-Levitov formula for free-fermions (appendix A). In order to illustrate this method we apply (25) to obtain new results in the classical hard rod gas and the quantum Lieb-Liniger model in the next sections.

## 5.2 Cumulants

Before we consider the specific models discussed above, we first show how to obtain cumulants from the expression (25). The first few cumulants provide important information about the shape of the distribution, and are the most accessible to numerical simulations and experiment; these are arguably the most important outcome of our expression for $F(\lambda)$.

The cumulants are evaluated from (25) via $c_k = \partial_\lambda^k F(\lambda)\Big|_{\lambda=0}$. The first cumulant is the average current, given in (17). For the second cumulant, omitting the $\theta$-dependence of the integrand for lightness of notation, we obtain

$$c_2 = \int \mathrm{d}\theta \, \rho_{\mathrm{p}} |v^{\mathrm{eff}}| (h_{i_*}^{\mathrm{dr}})^2 f, \tag{30}$$

where $\rho_{\mathrm{p}} = n(p')^{\mathrm{dr}}/(2\pi)$ is the quasiparticle density, and $f = -\mathrm{d}\log n/\mathrm{d}\epsilon$ is known as the statistical factor. The expression in (30) was already evaluated exactly in [32] by different methods, and follows as a consequence of current-current sum rules [101]. Our expression is in agreement with this, providing an important check on our result. The higher cumulants are new, and in particular (again omitting the explicit $\theta$-dependence of the integrand),

$$c_3 = \int \mathrm{d}\theta \, \rho_{\mathrm{p}} f \, |v^{\mathrm{eff}}| h_{i_*}^{\mathrm{dr}} \left[ (h_{i_*}^{\mathrm{dr}})^2 \tilde{f} s + 3 \left( (h_{i_*}^{\mathrm{dr}})^2 f s \right)^{\mathrm{dr}} \right], \tag{31}$$

where $\tilde{f} = -(\mathrm{d}\log f/\mathrm{d}\epsilon + 2f)$ and $s = \mathrm{sgn}(v^{\mathrm{eff}})$. We have also evaluated $c_4$, the result of which is presented in appendix B, but higher cumulants quickly become cumbersome. Details on the calculation of these quantities can also be found in appendix B. In [32], a natural linear-response formulation was also shown to reproduce $c_2$. The present results for $c_k$ agree with a generalisation of this linear-response formulation (see appendix C).

In the next section we confirm these exact predictions in the classical hard rod gas by direct numerical simulation, thereby verifying (25) while simultaneously obtaining new results for the paradigmatic hard rod gas.

## 6  Classical hard rod gas

In this section we use Monte Carlo simulations to verify the newly-found expressions for cumulants based on (25). This requires the specialisation of the above general expressions to the hard rod gas.

The hard rod gas is a one-dimensional classical system of rods of length $a$ whose whose only interactions are elastic collisions. Upon colliding, the rods swap velocities. The hydrodynamic description of the gas was derived rigorously in [78]. In our notations, $\mathsf{F}(\epsilon) = -e^{-\epsilon}$, $n(\theta) = e^{-\epsilon(\theta)}$, $f(\theta) = 1$, and the interactions are defined by $\varphi(\theta, \alpha) = -a$. In order to verify (25), we specialise to the hard rod gas and compare the predictions of the first four cumulants of the energy flow ($h_{i_*}(\theta) = \theta^2/2$) with direct Monte Carlo simulations of the gas. Using the exact TBA description [26], we first evaluate the predicted cumulants in a NESS from the partitioning protocol with initial left and right states that are thermal and boosted with different temperatures and boost velocities. See appendix D for a derivation of thermal distributions in the hard rod gas; these are normal distributions, proportional to $\exp(\beta(v - v_0)^2/2)$ where $v_0$ describes the boost velocity and $\beta$ the temperature of the bath. We then simulate the gas by running the (deterministic) hard rod dynamics from a sampled initial condition, where the initial left and right halves of the line are sampled with the prescribed left and right thermal boosted states. We add up the energy of the rods that pass through the centre of the system up to time $t$. This is done for multiple samples, from which we extract the cumulants and then scale by time. At large times, these numerical results are expected to agree with the cumulants evaluated in the NESS itself.[6]

---

[6]This is based on the fact that at large times, in the partitioning protocol, time-dependent correlation functions at the position $x = 0$ tend to their form in the state that is obtained on the ray $x = 0$. In integrable systems, this

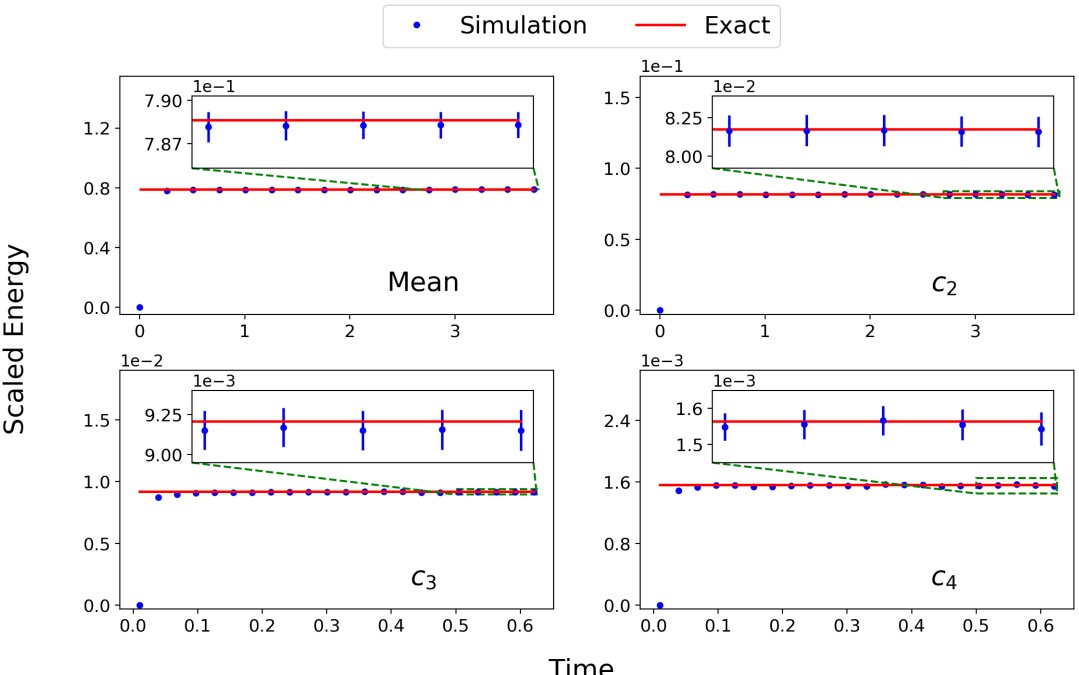

Figure 2: Insets highlight the agreement between theoretical predictions and Monte Carlo simulation of the first four energy flow cumulants in the hard rod gas. Theoretical predictions from (25) are the red lines, Monte Carlo results are the blue dots with error bars. The initial velocities of the rods are drawn from normal distributions, with mean 8 and variance 15, and mean $-3$ and variance 10, for the left and right system halves, respectively. Other parameters are: rod length $a = 0.56$; $10^5$ particles; $2 \times 10^7$ Monte Carlo samplings; initial system length $10^5$. Rod densities are fixed by the velocity variances in thermal distributions. The scaling parameter for the $y$-axis is $\bar{v}^3/a$ where $\bar{v}$ is the average rod speed. Error bars are found via bootstrap re-sampling using 3000 samples. Times plotted are chosen so as to reach the effective steady state before boundary effects arise; higher cumulants, which are affected by rarer events with faster moving rods, are sensitive to finite-size effects sooner.

The Monte Carlo error bars are obtained via the bootstrap sampling method which entails re-sampling with replacement from the obtained data set and calculating "alternative" values of the required cumulant [105]. The standard deviation of these values represents the associated uncertainty. Figure 2 shows cumulants of the steady state energy flow in the hard rod gas realised by Monte Carlo simulation, compared with results predicted from generalised hydrodynamics. It is clear that within error bars the prediction from (25) is successful – see appendix D for details on theoretical cumulant calculations in the hard rod gas, and appendix E for details on the Monte Carlo numerical simulation. Here, by boosting, the initial partitions are not just put into contact, but are thrust into each other. This is a highly non-trivial setup and the accurate prediction displays the power of the formalism employed here, providing strong evidence that (25) is correct.

---

fact can be observed, for generic observables, from the expressions for correlation functions in inhomogeneous states found in [34]. We note that in particular, the discussion in [34, sect 5.2.1] was inaccurate in claiming that the factor $V(\theta)$ would remain.

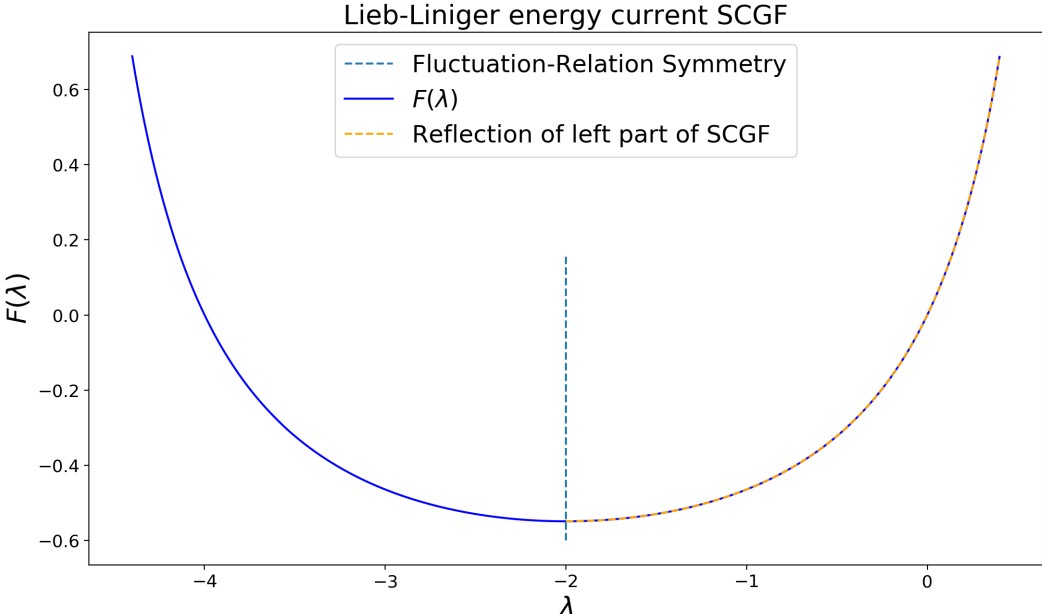

Figure 3: Numerical evidence for the fluctuation-relation symmetry in the Lieb-Liniger gas. Plot of the energy flow SCGF for the Lieb-Liniger model using (25). For this plot the parameters used are $\beta_l = 1$, $\beta_r = 5$, $c = 1$. The fluctuation-relation symmetry point is $(\beta_l - \beta_r)/2$, as indicated by the dotted line.

## 7 Quantum Lieb-Liniger gas

A different check on (25) is to confirm the expected non-equilibrium fluctuation relations (8). As mentioned at the end of section 2, general arguments suggest that the fluctuation relations should hold for the SCGF in the NESS from the partitioning protocol. We do not yet have an analytical proof of (8) from (25); this is nontrivial, as the fluctuation relations involve finite shifts of the generating parameter $\lambda$, while (25) is expressed in terms of the solution to a flow equation, which is local in $\lambda$. Nevertheless, one may verify the symmetry (8) by numerically solving the flow equation (24) and evaluating (25).

In order to provide verifications of our results beyond classical models, we choose to check the fluctuation relations for the energy current in the quantum Lieb-Liniger model. This model is important in the context of integrability as recent advances have rendered it accessible through cold atom experiments, see the book [77].

The Lieb-Liniger model describes a one-dimensional gas of Bose particles with $\delta$-function interactions. Specialising Eq. (25) to this model, we obtain explicit predictions for all the large-scale cumulants for transport, including for the total number of particles transferred ($h_{i_*}(\theta) = 1$) and the total energy ($h_{i_*}(\theta) = \theta^2/2$). Again we analyse the energy SCGF in the partitioning protocol NESS (using its exact TBA description [21]) with initial states set by different purely thermal baths, at inverse temperatures $\beta_l$ and $\beta_r$. We set $\mathsf{F}(\epsilon) = -\log(1+e^{-\epsilon})$, $n(\theta) = 1/1+e^{\epsilon(\theta)}$, and $f = 1-n$, while interactions are defined by $\varphi(\theta, \alpha) = 4c/((\theta-\alpha)^2+4c^2)$ where $c$ is the coupling strength. The equation (24) is solved numerically using an iterative approach known as Picard's method [106]. The SCGF is plotted in fig. 3; it is convex as it should be [42], and grows sharply near the values $-\beta_r$ and $\beta_l$. These are the values at which divergences in the SCGF occur in free bosonic models (and also in conformal field theory [57]), thus suggesting that at these values, the free bosonic physics of the Lieb-Liniger model domi-

nates. The plot also indicates that the energy flow in the Lieb-Liniger model, with these initial conditions, satisfies the non-equilibrium fluctuation relations (8). Appendix F provides more extensive numerical verifications of the fluctuation relations by exploring a range of parameters in the Lieb-Liniger model and shows numerical evidence in the hard rod gas too.

The validity of the fluctuation relations in the Lieb-Liniger model and hard rod gas provides further support both for (25), and for the general consistency of the theory of non-equilibrium transport.

# 8 Conclusion

In this paper, we have obtained the exact SCGF, or full counting statistics, for transport of arbitrary conserved quantities in a wide family of integrable models, and in arbitrary GGEs, including non-equilibrium steady states. The results have been obtained by combining recent developments in integrable systems in the context of generalised hydrodynamics [21, 22], in particular the flux Jacobian obtained in [32], with a new ballistic fluctuation formalism developed in [92]. They significantly generalise previous expressions and studies in free particle models [50–55] and in one-dimensional conformal field theory [57–59]. To our knowledge, these are the first exact results for transport SCGFs in interacting homogeneous integrable systems, and provide an entirely new application of the hydrodynamic theory of integrable systems. The SCGF allows one to extract, order by order, exact expressions for the scaled transport cumulants purely in terms of quantities from the thermodynamic Bethe ansatz. We have provided explicit expressions for the first few cumulants. The results can be immediately applied, for instance, to the paradigmatic classical hard rod gas, the quantum Lieb-Liniger model, as well as other classical and quantum chains and field theories such as the sinh-Gordon and sine-Gordon model and the Heisenberg chain, and to soliton gases [80–84]. Importantly, we have explicitly verified the validity of the formalism by comparing the exact formulae for cumulants with Monte Carlo simulations in the classical hard rod gas. We have also explained how to apply the formalism to the quantum Lieb-Liniger gas, and we have checked, by numerically evaluating the exact expression, that the SCGF satisfies the non-equilibrium fluctuation relations both in the classical hard rod and quantum Lieb-Liniger gases.

Many questions arise from the present results. First, a more in-depth analysis of the SCGF and its properties would be very interesting, including an analytic proof of the fluctuation relations. It would particularly useful if simplifications of the general formula (25) could be found, allowing for a better analysis of the SCGF at large or complex values of the generating parameter $\lambda$. For instance, the analytic structure of the SCGF in the complex $\lambda$-plane would provide information about the structure of transport degrees of freedom, see [107]. Applications to other integrable models would be very desirable. As our main results, even though accurate, are not derived in a mathematically rigorous fashion, it is paramount to have comparisons with tDMRG studies for quantum models. Furthermore, as explained in [92], the SCGF gives the exact exponential decay of dynamical two-point correlation functions of twist fields; the latter, in many cases, represent order parameters, and examples are the exponential fields in the sine-Gordon model. It would be useful to verify the predicted exponents, and further study their consequences. Finally, it would be interesting to explore the empirical counting statistics of the Lieb-Liniger gas in experiments on cold atomic gases, and compare them with our theoretical results. Tantalisingly, generalisations to inhomogeneous non-stationary situations might also be possible with current technology within GHD.

## Acknowledgments

We thank Takato Yoshimura for many discussions on the subject of this paper at the early stages of the project, as well as A. Abanov, Dinh-Long Vu, Juan P. Garrahan, Herbert Spohn and Tomohiro Sasamoto for discussions and comments. JM acknowledges funding from the EPSRC Centre for Doctoral Training in Cross-Disciplinary Approaches to Non-Equilibrium Systems (CANES) under grant EP/L015854/1. RJH thanks the London Mathematical Laboratory for research support in the form of an External Fellowship. BD is Royal Society Leverhulme Trust Senior Research Fellow, ref. SRF\R1\180103. BD is grateful to École Normale Supérieure de Paris for an invited professorship (2018), where part of this work was carried out. MJB and BD acknowledge hospitality and funding from the Erwin Schrödinger Institute in Vienna (Austria), and BD acknowledges hospitality and funding from the Galileo Galilei Institute in Florence (Italy). BD's research was supported in part by Perimeter Institute for Theoretical Physics. Research at Perimeter Institute is supported by the Government of Canada through the Department of Innovation, Science and Economic Development and by the Province of Ontario through the Ministry of Research, Innovation and Science. All authors thank the Centre for Non-Equilibrium Science (CANES) and the Thomas Young Centre (TYC).

## A  Specialisation to the Lesovik-Levitov formula

In free-fermion models, there is a well-known formula for the SCGF for particle transfer through an impurity between two "leads", the Lesovik-Levitov formula [50]. The Lesovik-Levitov formula specialised to pure transmission (that is, without an impurity) should agree with our formula (25), specialised to free-fermionic particles, to the steady state arising from an initial imbalance in the partitioning protocol, and to the study of particle transfer. Here we verify this in a very simple example, corresponding to the choice of free massless, charge-less fermionic leads, which have linear dispersion relation. In this case, the Lesovik-Levitov formula takes the form [52]

$$F(\lambda) = \int_{-\infty}^{\infty} \frac{d\omega}{2\pi} \log\left[1 - n_l(\omega)(1 - n_r(\omega))(1 - e^{\lambda}) - n_r(\omega)(1 - n_l(\omega))(1 - e^{-\lambda})\right], \quad (32)$$

where $n_j(\omega)$ is the Fermi occupation function for the initial left, $j = l$, and right, $j = r$, reservoirs; for instance, with temperatures $T_j$ and chemical potentials $\mu_j$", we have,

$$n_j(\omega) = \frac{1}{e^{(\omega - \mu_j)/T_j} + 1}. \quad (33)$$

Here $\omega$ plays the role of an energy; it is not bounded from below because of the linear dispersion relation.

In our formalism, we have two particle types, $\sigma = \pm 1$, corresponding to the right-movers and left-movers of the massless free-fermion theory. We may choose momentum $p(\theta, \sigma) = \theta$; the energy function takes the form $E(\theta, \sigma) = \sigma\theta$ so that the velocity is $v(\theta, \sigma) = \sigma$. The theory is free, hence this also equals the effective velocity, and the dressing operation is trivial. The non-equilibrium steady state in free-fermion models has been known exactly for some time [67, 68], and, in our notations, has pseudoenergy given by

$$\epsilon(\theta, \sigma; \lambda) = \frac{1}{T_l}(\theta - \mu_l)\Theta(\sigma) + \frac{1}{T_r}(-\theta - \mu_r)\Theta(-\sigma). \quad (34)$$

This embodies the independent thermalisation of right- and left-movers with respect to the

initial left and right states, respectively. Solving (24) we find

$$\epsilon(\theta,\sigma;\lambda) = \left[\frac{1}{T_l}(\theta - \mu_l) - \lambda\right]\Theta(\sigma) + \left[\frac{1}{T_r}(-\theta - \mu_r) + \lambda\right]\Theta(-\sigma) \tag{35}$$

and (25) becomes

$$F(\lambda) = \int_{-\infty}^{\infty}\frac{d\theta}{2\pi}\sum_{\sigma}\left(\log(1 + e^{-\epsilon(\theta,\sigma;\lambda)}) - \log(1 + e^{-\epsilon(\theta,\sigma;0)})\right). \tag{36}$$

Changing variable to $\omega = \sigma\theta$, followed by some simple algebraic manipulations, it can be seen this agrees with (32).

# B   Calculating $c_3$ and $c_4$

The cumulants are obtained from the SCGF (25) as $c_n = \partial_\lambda^n F(\lambda)\big|_{\lambda=0}$; this appendix outlines the steps involved in taking the $\lambda$-derivatives.

This appendix makes use of a flow equation on the state $n(\theta;\lambda)$. To obtain this expression we use $f = -d\log n/d\epsilon$ with the flow equation (24) to get

$$\partial_\lambda n(\theta;\lambda) = \text{sgn}\left(v^{\text{eff}}(\theta;\lambda)\right)h_{i_*}^{\text{dr}}(\theta;\lambda)n(\theta;\lambda)f(\theta;\lambda). \tag{37}$$

Calculating $\lambda$-derivatives of (25) requires the following identities, gleaned from understanding the integral-operator structure of the dressing operator:

$$\partial_\lambda X^{\text{dr}}(\theta;\lambda) = (sfh^{\text{dr}}X^{\text{dr}})^{\text{dr}}(\theta;\lambda) - f(\theta;\lambda)s(\theta;\lambda)h^{\text{dr}}(\theta;\lambda)X^{\text{dr}}(\theta;\lambda), \tag{38}$$

$$\int d\theta\, n(\theta)X(\theta)Y^{\text{dr}}(\theta) = \int d\theta\, n(\theta)X^{\text{dr}}(\theta)Y(\theta), \tag{39}$$

where $s(\theta;\lambda) = \text{sgn}\left(v^{\text{eff}}(\theta;\lambda)\right)$, and $X(\theta)$ and $Y(\theta)$ stand for any two quantities within the GHD description.

Going forward we use the $\lambda$-dependent state $n(\theta;\lambda)$ which is defined by (37). The $\lambda$-dependent current is obtained from the expression

$$\langle j(\lambda)\rangle = \int\frac{d\theta}{2\pi}E'(\theta)n(\theta;\lambda)h_{i_*}^{\text{dr}}(\theta;\lambda). \tag{40}$$

At $\lambda = 0$ this correctly produces the steady-state current as given by the first expression in (17).

In order to ensure the next calculations are more readable, the $\theta$-dependence in the notation is suppressed with the understanding that all terms inside the $\theta$ integrals are $\theta$ dependent. Furthermore, the following simplified notation is introduced: $s(\lambda) \equiv \text{sgn}\left(v^{\text{eff}}(\theta;\lambda)\right)$ and $H(\lambda) \equiv h_{i_*}^{\text{dr}}(\theta;\lambda)$.

The second cumulant $c_2$ is found by taking a $\lambda$-derivative of the $\lambda$-dependent current and setting $\lambda = 0$,

$$\partial_\lambda\langle j(\lambda)\rangle = \int\frac{d\theta}{2\pi}E'[\partial_\lambda n(\lambda)H(\lambda) + n(\lambda)\partial_\lambda H(\lambda)]$$

$$= \int\frac{d\theta}{2\pi}E'[s(\lambda)H^2(\lambda)n(\lambda)f(\lambda) + n(\lambda)(sfH^2)^{\text{dr}}(\lambda) - n(\lambda)f(\lambda)s(\lambda)H^2(\lambda)]$$

$$= \int\frac{d\theta}{2\pi}(E')^{\text{dr}}(\lambda)n(\lambda)s(\lambda)f(\lambda)H^2(\lambda), \tag{41}$$

where in the second line we used (37) and (38) while the third line required (39). At $\lambda = 0$ this correctly reproduces $c_2$ from the literature [32].

For higher-order derivatives special care of the terms $\partial_\lambda s(\lambda)$ is necessary. Recall $s(\lambda)$ is a sign function and the derivative of this produces a $\delta$-function. The $\delta$-function leads to terms evaluated at $\theta^*(\lambda)$ where $v^{\text{eff}}(\theta^*(\lambda); \lambda) = 0$. This can be problematic, as in the partitioning protocol considered in this work, $n(\theta^*(\lambda))$ is ill-defined. However, for $c_3$ the ambiguity resolves fairly

straightforwardly. On taking the derivative of $\partial_\lambda \langle j(\lambda) \rangle$ there is a term that contains $\partial_\lambda \text{sgn}\big(v^{\text{eff}}(\theta; \lambda)\big) = \delta(v^{\text{eff}}(\theta; \lambda))\partial_\lambda v^{\text{eff}}(\theta; \lambda)$. The trick comes from recalling from the definition of $v^{\text{eff}}$ in section 4 that $(E')^{\text{dr}}(\theta; \lambda) = v^{\text{eff}}(\theta; \lambda)(p')^{\text{dr}}(\theta; \lambda)$. Thus we have a term $\int \frac{d\theta}{2\pi} (p')^{\text{dr}}(\theta; \lambda)v^{\text{eff}}(\theta; \lambda)\delta(v^{\text{eff}}(\theta; \lambda))\partial_\lambda v^{\text{eff}}(\theta; \lambda)n(\lambda)f(\lambda)H^2(\lambda)$. The $\delta$-function sets $v^{\text{eff}} = 0$ which ensures we do not need to evaluate $n(\theta^*(\lambda))$. The remaining terms all follow from use of (37), (38) and (39) which leads to

$$
\begin{aligned}
\partial_\lambda^2 \langle j(\lambda) \rangle = \int \frac{d\theta}{2\pi} \Big[ & (sf(E')^{\text{dr}}H)^{\text{dr}}(\lambda)n(\lambda)s(\lambda)f(\lambda)H^2(\lambda) \\
& + 2(E')^{\text{dr}}(\lambda)n(\lambda)s(\lambda)f(\lambda)H(\lambda)(sfH^2)^{\text{dr}}(\lambda) + (E')^{\text{dr}}(\lambda)n(\lambda)f(\lambda)H^3(\lambda)\tilde{f}(\lambda) \Big] \\
= \int \frac{d\theta}{2\pi} & (E')^{\text{dr}}(\lambda)n(\lambda)s(\lambda)f(\lambda)H(\lambda)\big[ s(\lambda)H^2(\lambda)\tilde{f} + 3(sfH^2)^{\text{dr}}(\lambda) \big],
\end{aligned}
\tag{42}
$$

where $\tilde{f} = -(d\log f/d\epsilon + 2f)$ and $s^2(\lambda) = 1$ was used throughout. The final expression (31) for $c_3$ is obtained by recalling again the identities $(E')^{\text{dr}} = v^{\text{eff}}(p')^{\text{dr}}$, $|v^{\text{eff}}| = v^{\text{eff}} \text{sgn}(v^{\text{eff}})$, and $\rho_p = n(p')^{\text{dr}}/(2\pi)$ before finally setting $\lambda = 0$.

Although the results are exact, one can see the increasing complexity of the required manipulations for higher cumulants. To obtain $c_4$ involves the same steps as above, first using (37) and (38) followed by acting on the term gained from $\partial_\lambda (E')^{\text{dr}}$ with (39). However, a further complication arises when considering the term $\partial_\lambda (sfH^2)^{\text{dr}}(\lambda)$. We now explain the specific issue and how to overcome it but we do not repeat manipulations already covered in the calculations of $c_2$ and $c_3$..

In order to calculate $\partial_\lambda (sfH^2)^{\text{dr}}(\lambda)$, consider the integral representation of a dressed object. From (18), the dressing operator is $h^{\text{dr}}(\theta) = h(\theta) + \int \frac{d\alpha}{2\pi} \varphi(\alpha, \theta)n(\alpha)h^{\text{dr}}(\alpha)$. With this definition

$$
\begin{aligned}
\partial_\lambda (sfH^2)^{\text{dr}}(\lambda) = & \partial_\lambda(s(\theta; \lambda)f(\theta; \lambda)H^2(\theta; \lambda)) \\
& + \partial_\lambda \int \frac{d\alpha}{2\pi} \varphi(\alpha, \theta)n(\alpha; \lambda)s(\alpha; \lambda)f(\alpha; \lambda)H(\alpha; \lambda)^2 \\
& + \partial_\lambda \int \frac{d\alpha}{2\pi} \varphi(\alpha, \theta)n(\alpha; \lambda) \int \frac{d\alpha'}{2\pi} \varphi(\alpha', \alpha)n(\alpha'; \lambda)s(\alpha'; \lambda)f(\alpha'; \lambda)H(\alpha'; \lambda)^2 \\
& + \dots,
\end{aligned}
\tag{43}
$$

where we have displayed only the first three terms of the infinite sequence arising from the iterative equation defined by the dressing operator. Everything is fairly straightforward except that the derivatives of $s(\theta; \lambda)$ do not resolve the issues of evaluating $n(\theta^*(\lambda))$ as in $c_3$ before. To get around this issue, consider splitting the integrals above such that $\int d\theta s(\theta^*(\lambda)) = -\int_{-\infty}^{\theta^*(\lambda)} d\theta + \int_{\theta^*(\lambda)}^{\infty} d\theta$. Then using the Leibniz integral rule, the boundary terms which come from $\partial_\lambda \int d\theta s(\theta^*(\lambda))$ cancel each other out. This removes the ambiguity from (43), as is to be expected since the sign function entered our initial calculations as a shorthand. In fact, it is possible to do all calculations without the explicit use of this function, instead computing under split integrals from the start so that there is never a question of

ambiguity. With this issue resolved we write

$$\partial_\lambda (sfH^2)^{\mathrm{dr}}(\lambda) = 3\Big(fsH(fsH^2)^{\mathrm{dr}}\Big)^{\mathrm{dr}}(\lambda) - f(\lambda)s(\lambda)H(\lambda)(fsH^2)^{\mathrm{dr}}(\lambda) + (f\tilde{f}H^3)^{\mathrm{dr}}(\lambda). \quad (44)$$

The rest of the calculation, although tedious, follows the same principles as before. For comparative purposes we write $c_4$ in the more formal notation used in (31) for $c_3$:

$$c_4 = \int \mathrm{d}\theta\, \rho_{\mathrm{p}} f\, v^{\mathrm{eff}} \Big\{ (h^{\mathrm{dr}}_{j_*})^4 s\hat{f}\tilde{f} + 3s((sf(h^{\mathrm{dr}}_{j_*})^2)^{\mathrm{dr}})^2 + 4h^{\mathrm{dr}}_{j_*} s(f\tilde{f}(h^{\mathrm{dr}}_{j_*})^3)^{\mathrm{dr}} + 6\tilde{f}(h^{\mathrm{dr}}_{j_*})^2 (sf(h^{\mathrm{dr}}_{j_*})^2)^{\mathrm{dr}}$$

$$+ 12h^{\mathrm{dr}}_{j_*} s(sfh^{\mathrm{dr}}_{j_*}(sf(h^{\mathrm{dr}}_{j_*})^2)^{\mathrm{dr}})^{\mathrm{dr}} \Big\}, \quad (45)$$

where $\hat{f} = -(\mathrm{d}\log(f\tilde{f})/\mathrm{d}\epsilon + 3f)$.

## C Cumulants from a linear response principle

We show that the cumulants obtained by taking $\lambda$-derivatives of (25), can also be obtained by following a linear response principle, generalising the linear-response formulation of the second cumulant found in [32].

The linear response formulation for the cumulants, in a given state of the system, is obtained by considering a certain partitioning protocol with respect to that state. Specifically, the required partitioning protocol is that whose initial condition, on the left/right, is the system state perturbed by $\pm(\mu/2)Q$, where $Q$ is the conserved charge of interest. One then looks at the NESS emerging from this partitioning protocol. Here we show that taking the $\mu$-derivative of the occupation function representing this NESS, evaluated at $\mu = 0$, results in the same expression as that obtained by taking the $\lambda$-derivative at $\lambda = 0$. By recursively applying this procedure, one can then obtain all $\lambda$ derivatives.

Recall the expression for the TBA pseudoenergy (16):

$$\epsilon(\theta) = w(\theta) + \int \frac{\mathrm{d}\alpha}{2\pi}\, \varphi(\theta, \alpha) \mathsf{F}(\epsilon(\alpha)), \quad (46)$$

where $w(\theta)$ is a source term defining the initial state – in (16) we used $w(\theta) = \sum_i \beta^i h_i(\theta)$).

As mentioned, the basis of the linear-response formulation of the cumulants in a particular state with occupation function $n(\theta)$, is to first evaluate the occupation function for the NESS emerging from the partitioning protocol with initial condition where both halves are set to the state under consideration, but with a "perturbation" by $\pm(\mu/2)Q$ in the left and right GGEs, respectively. The NESS occupation function, solution to this partitioning protocol, is given in [21, 22], and will be denoted $[n]_\mu(\theta)$. It takes the form $[n]_\mu(\theta) = n_{l;\mu}(\theta)\Theta(v^{\mathrm{eff}}_\mu(\theta)) + n_{r;\mu}(\theta)\Theta(-v^{\mathrm{eff}}_\mu(\theta))$ where $n_{l,r;\mu}(\theta)$ are constructed from the modified source terms of the pseudoenergy corresponding to the initial state of the protocol, $w_{l,r;\mu}(\theta) = w(\theta) \pm (\mu/2)h(\theta)$, and $v^{\mathrm{eff}}_\mu(\theta)$ is evaluated in the state $[n]_\mu(\theta)$.

Consider the $\mu$-derivative of $[n]_\mu(\theta)$. We may recast the form of the solution $[n]_\mu(\theta)$ in terms of the pseudoenergy $[\epsilon]_\mu(\theta)$, as they are related in a simple algebraic way. Clearly, $\partial_\mu [n]_\mu(\theta) = (\partial_\epsilon n)_{\epsilon = [\epsilon]_\mu(\theta)} \partial_\mu [\epsilon]_\mu(\theta)$. The first factor was presented in the main text: $\partial_\epsilon n = -nf$. By the linear-response construction and using the definition of the dressing operation, the second factor is $\partial_\mu [\epsilon]_\mu(\theta) = \partial_\mu \epsilon_{l;\mu}(\theta)$ if $v^{\mathrm{eff}}_\mu(\theta) > 0$, and $\partial_\mu [\epsilon]_\mu(\theta) = \partial_\mu \epsilon_{r;\mu}(\theta)$ if $v^{\mathrm{eff}}_\mu(\theta) < 0$, with a delta-function contribution at $v^{\mathrm{eff}}_\mu(\theta) = 0$. Using $\partial_\mu \epsilon_{l,r;\mu} = \pm h^{\mathrm{dr}}_{l,r;\mu}$ and the fact that $h^{\mathrm{dr}}_{l,r;0} = h^{\mathrm{dr}}$, the delta-function contribution vanishes at $\mu = 0$. We are left with

$$\partial_\mu [n]_\mu(\theta)|_{\mu=0} = h^{\mathrm{dr}}(\theta) f(\theta) n(\theta)\, \mathrm{sgn}(v^{\mathrm{eff}}(\theta)). \quad (47)$$

Comparing this equation to (37), which is a re-expression of (24), we observe that

$$\partial_\mu[n]_\mu(\theta)|_{\mu=0} = \partial_\lambda n(\theta;\lambda)|_{\lambda=0}. \tag{48}$$

The equality does not hold in general when the derivatives are evaluated at $\mu \neq 0$ and $\lambda \neq 0$. However, since the equality holds for every state, we can follow the same procedure starting with the $\lambda$-dependent state $n(\theta;\lambda)$, and considering a $\mu$-modification of it. We obtain

$$\partial_\mu[n(\cdot;\lambda)]_\mu(\theta)\big|_{\mu=0} = \partial_\lambda n(\theta;\lambda). \tag{49}$$

In short, the method outlined here gives an alternative way of expressing the $\lambda$-flow: the first-order variation of the state in $\lambda$ is equated with its first-order variation in *another parameter*, $\mu$, obtained by first applying an inhomogeneous perturbation with opposite chemical potentials $\pm\mu Q/2$ on the left and right halves – a bias – to the state $n(\cdot;\lambda)$, and then letting the state evolve for an infinite time towards the emerging steady state, getting $[n(\cdot;\lambda)]_\mu$. Again, the equality holds only for first variations; there is in general no equality for second and higher derivatives (except for free-particle models), and no group property, $[[n]_\mu]_{\mu'} \neq [n]_{\mu+\mu'}$. Thus, this approach does not provide a finite-$\lambda$ solution to the flow, but a re-writing of the flow equation.

The full re-writing is as follows. Recall from the previous appendix that (37) – specifying $\lambda$-derivatives – can be used to find the cumulants from (25). Since (49) states that the linear response in $\mu$ (that is, the $\mu$-derivative at $\mu = 0$), reproduces the first $\lambda$-derivative, this implies that linear response can be used to reproduce all cumulants. Explicitly, writing $n(\theta;\lambda) = n^{(0)}(\theta) + \lambda n^{(1)}(\theta) + (\lambda^2/2)n^{(2)}(\theta) + \ldots$, we determine the $n^{(i)}(\theta)$'s by solving

$$n^{(1)}(\theta) + \lambda n^{(2)}(\theta) + \ldots = \partial_\mu\big[n^{(0)} + \lambda n^{(1)} + (\lambda^2/2)n^{(2)} + \ldots\big]_\mu(\theta)|_{\mu=0} \tag{50}$$

recursively in powers of $\lambda$.

Via the above procedure, linear response is capable of exploring the space close to any fixed $\lambda$ via a small shift in the initial charge, and hence can be used, in principle, to calculate all cumulants.

## D   The hard rod gas and its thermal distribution

The hard rod (HR) gas provides a simple model in which to test our results. We here explain how to generate the inputs for the theoretical predictions of $c_1, c_2, c_3$ and $c_4$ plotted in fig. 2.

To obtain the theoretical values for the cumulants, the following are required: the conserved quantity $h(\theta)$; the occupation function $n(\theta)$ together with the related pseudoenergy $\epsilon(\theta)$ and particle density $\rho_p(\theta)$, the dressing operation, the effective velocity $v^{\text{eff}}(\theta)$, and the statistical factor $f(\theta)$. Here $\theta$ is the velocity and we take unit mass.

As stated in the main text, in the HR gas the differential scattering amplitude is given by $\varphi(\theta, \alpha) = -a$ where $a$ is the rod length. The constant interaction term simplifies the dressing operation (see again the main text), giving

$$h^{\text{dr}}(\theta) = h(\theta) - \frac{a}{1+ab}\int \frac{d\alpha}{2\pi} n(\alpha)h(\alpha), \tag{51}$$

with

$$b = \int \frac{d\alpha}{2\pi} n(\alpha). \tag{52}$$

This produces further simplifications of $v^{\text{eff}}(\theta)$ since $v^{\text{eff}}(\theta) = (E')^{\text{dr}}(\theta)/(p')^{\text{dr}}(\theta)$. Further, in the HR gas $f(\theta; \lambda) = 1$ (independent of the state) and

$$n(\theta) = e^{-\epsilon(\theta)}, \tag{53}$$

as this is a gas of classical particles with free energy function $\mathsf{F}(\epsilon) = e^{-\epsilon}$. The particle density is given by $\rho_p(\theta) = n(\theta)(p')^{\text{dr}}((\theta))/(2\pi)$ which, using $p'(\theta) = 1$ and (51), is easily shown to yield

$$\rho_p(\theta) = \frac{1 - a\rho}{2\pi} n(\theta), \quad \text{where } \rho = \int d\alpha \, \rho_p(\alpha) = \frac{b}{1 + ab}. \tag{54}$$

Since the differential scattering phase is constant in the HR gas, the pseudoenergy can be expressed as $\epsilon(\theta) = w(\theta) + z$, where the constant $z$ satisfies

$$z = ade^{-z}, \qquad d = \int \frac{d\alpha}{2\pi} e^{-w(\alpha)}. \tag{55}$$

The equation for $z$ is solved using the Lambert-W function as $z = W(ad)$.

For the particular situation of interest, we are concerned with energy currents so $h(\theta)$ takes the simple form $\theta^2/2$. For the HR gas in the steady state arising from the partitioning protocol, a result of [26] provides the exact expression for $n(\theta)$ in terms of the occupation functions $n_l(\theta)$ and $n_r(\theta)$ in the initial left and right baths of the protocol. Here, the initial state is fixed using two boosted thermal distributions. In order to apply the result of [26], we therefore only need to describe what form $n(\theta)$ takes for thermal distributions in the HR gas; Galilean boosting is simply a shift of $\theta$. To obtain a thermal distribution in the HR gas, one may naively assume that fixing the source term $w(\theta)$ of the pseudoenergy (46) to be Gaussian is sufficient. However, we show that for truly thermal distributions, the starting rod density per unit length must also be fixed in a particular way.

To fix a thermal state, we choose a thermal source term defined by $w^{(\text{th})}(\theta) = \beta\theta^2/2$. It is now clear that setting a thermal source term will effect the particle density. The thermal $d^{(\text{th})}$ is a Gaussian integral giving $d^{(\text{th})} = 1/\sqrt{2\pi\beta}$. This is used to find $\epsilon^{(\text{th})} = \beta\theta^2/2 + W(ad^{(\text{th})})$. Then, using $n(\theta) = e^{-\epsilon(\theta)}$ with (54),

$$\rho_p^{(\text{th})} = \frac{e^{-\epsilon^{(\text{th})}(\theta)}}{2\pi(1 + ab^{(\text{th})})} = \frac{e^{-\beta\theta^2/2}e^{-W(ad^{(\text{th})})}}{2\pi(1 + W(ad^{(\text{th})}))}. \tag{56}$$

This is how the initial thermal densities are constructed for the Monte Carlo simulation of the hard rod gas. Since we investigate boosted thermal distributions, we use such Gaussian distributions with non-zero means; the above details remain unaffected other than in the final equality $\theta \to \theta - \mu$.

## E    Monte Carlo details

We here describe in detail the Monte Carlo procedure used to obtain $c_2$, $c_3$ and $c_4$ for the HR gas in the partitioning protocol used in fig. 2. In the partitioning protocol the system is split into two halves defined by different boosted thermal distributions for the left and right side of the partition (see appendix D above). This defines both the rod velocities, through sampling from a Gaussian distribution, and the rod densities, through (56). On the left side of the partition we used a Gaussian with a mean of 8 and standard deviation of 15, on the right a mean of $-3$ and standard deviation of 10. The initial length scale in the system is defined by the distance between the leftmost rod of the left partition and the rightmost rod of the right

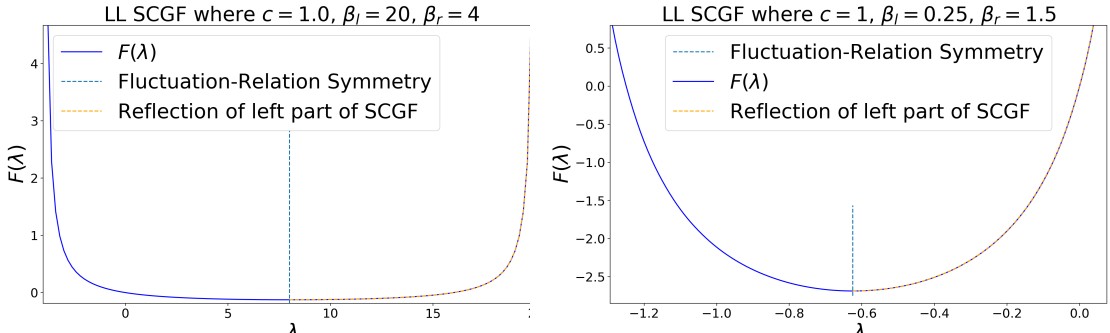

Figure 4: Energy flow SCGF for the Lieb-Liniger model using equation (25). Both plots use an interaction strength of $c = 1$. In (a) the parameters used are $\beta_l = 20$, $\beta_r = 4$ and in (b) $\beta_l = 0.25$, $\beta_r = 1.5$. In both cases a dotted line is plotted for the expected symmetry point at $(\beta_l - \beta_r)/2$. Both plots also indicate the symmetry by reflecting the left-hand side of the plot onto the right-hand side of the plot.

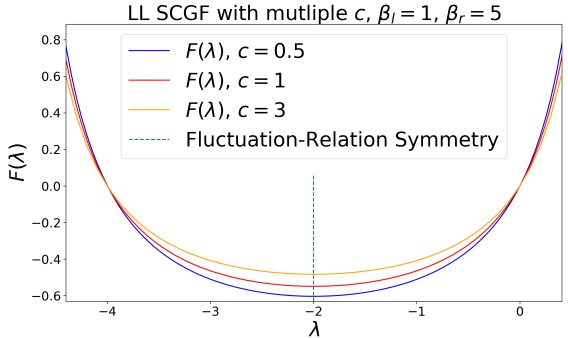

Figure 5: Same as fig. 4 but with $\beta_l = 1$ $\beta_r = 5$ and various values of $c$. The FR symmetry point, $(\beta_l - \beta_r)/2$, is again shown by the dotted line.

partition. To ensure we study a system where interactions are important while also avoiding packing the rods too densely, we enforce the initial length scale to be half-populated by rods, taking into account rod lengths. It is easy to show that the initial length scale and the rod length are uniquely determined by the choice of rod number, left/right Gaussian parameters and the constraint the initial length scale is half-filled with rods. We used $10^5$ rods in our simulation. We stress again that all the stochasticity is contained within the initial conditions as the initial rod velocities are randomly drawn from Gaussian distributions but, the time evolution is deterministic. For a given realisation of initial velocities, we count the total amount of energy that passes through the midpoint of the system during a long time interval $t$. To gain statistics on the energy flow, we allow multiple realisations of initial rod velocities and record the total energy flow for each. From the data collected the scaled cumulants can be determined.

## F  Numerical evidence for fluctuation relations

We provide further numerical evidence that the SCGF given in (25) satisfies fluctuation relations (FRs). Recall from (8), and the discussion below it, that in our setup FRs take the form

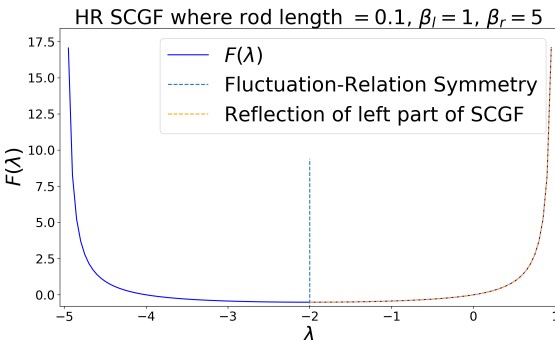

Figure 6: Plot of the energy flow SCGF for the hard rod model using equation (25). For this plot the parameters used are $\beta_l = 1$, $\beta_r = 5$ and rod length $= 0.1$. Once again the dotted line is the expected FR symmetry point and the curve is reflected to indicate how symmetrical it is.

$F(\lambda) = F(\beta_l - \beta_r - \lambda)$ where $\beta_l$ and $\beta_r$ are the left and right inverse temperatures in the partitioning protocol. Thus the FRs are exposed by a symmetry in the plot of the SCGF about the point $(\beta_l - \beta_r)/2$. We stress that the figures in this section are not produced via Monte Carlo simulations but rather represent numerical solutions for (25) for various model scenarios. In fig. 4 we plot the SCGF for the Lieb-Liniger model with different temperatures but the same interaction strength $c$. fig. 5 displays the results of varying the interaction strength while maintaining the same temperatures. In order to be complete, fig. 6 shows HR-specific results where we use similar parameters as for the previous plots. In all cases the symmetry is prominent which provides strong numerical evidence that our exact SCGF formula does indeed satisfy FRs.

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
