# Peer review of "Transport fluctuations in integrable models out of equilibrium"

_SciPost Physics, doi:SciPost Phys. 8, 007 (2020)_

## Round 3 · Referee Report · Anonymous · 2019-7-24

Strengths

1. Timely and interesting results
2. Predictions are tested against microscopic classical simulations

Weaknesses

1. Some aspects of presentation

Report

The authors extend the recently developed hydrodynamics of integrable models to allow for the computation of higher cumulants of currents. Their key result is a new analytical prediction (eq. 25) for the scaled cumulant generating function. Predictions obtained from this formula are tested directly against Monte Carlo simulations of the classical hard-rod gas, showing good agreement for the first four cumulants. The authors further check that their formula is consistent with known analytical results for free fermions and the “partitioning protocol” in the Lieb-Liniger gas.
The results presented are timely and interesting, building valuable new connections between generalized hydrodynamics and large-deviation theory. My only criticisms of the paper are some minor issues of presentation, that I would ask the authors to reconsider before publication:
1) The word “exact” is used to describe results ranging from rigorous to conjectural, including the main result, eq. 25. However, to the best of my knowledge, there is no published, generally valid, proof of the fundamental assumption, eq. 17, even though it is believed to hold for many integrable models. If the authors will refer to “exact results”, they ought to address these nuances.
2) The main result is derived for homogeneous equilibrium states, but the example studied in the paper is an inhomogeneous problem. Of course, I understand that in a particular scaling limit, along a particular ray, the partitioning protocol generates homogeneous states, but this is never clearly stated in the paper. Instead, sentences in the manuscript like “GGE states include NESSs” confuse the issue.
If the authors can address these minor points of presentation, I fully recommend the manuscript for publication.

Requested changes

1. Sec 1, p. 2: it is stated “GHD…gives rise to a panoply of exact results, including exact nonequilibrium flows [21, 22, 26–29], Drude weights [30–33] and large-scale correlations [34, 35], as well as a first-principles theoretical solution to the quantum Newton’s cradle set-up at arbitrary coupling strength” While there is no doubt that GHD is a successful theory, I find this sentence misleading. Many of the results cited are well-tested conjectures rather than “exact results”. Moreover, the quantum Newton’s cradle is a difficult open problem, for which GHD yields a simplified model rather than a “theoretical solution”.
2. Sec 1, p. 3-4, also Sec 5 and Sec 7: the authors refer repeatedly to their eq. 25 as “exact”. As mentioned above, this is conditional on exactness of eq. 17, which is conjectural in general.
3. Sec 2, p. 5: It is stated that the results of the paper apply to homogeneous GGEs. I do not understand the subsequent remark that “GGE states include NESSs” – without further qualification, this seems incorrect.
4. Sec 2, p.5, also Sec 7: In Sec 2, it is stated that "NESSs emerging from the partitioning protocol were constructed exactly in integrable models in [21, 22, 26]". For quantum models the hydrodynamic NESS is still a conjecture, and not "constructed exactly" (as was achieved in Ref [26] for hard rods).
5. Sec 8, p. 16: “As our main results, even though completely accurate, are derived in a mathematically rigorous fashion, it is paramount to have comparisons with tDMRG studies for quantum models” I find this sentence inappropriate, as the assertions “completely accurate” and “mathematically rigorous” add no scientific value. If the derivation were truly rigorous, further tests against DMRG would be redundant.

---

## Round 3 · Referee Report · Anonymous · 2019-7-30

Strengths

1) New results valid for generic interacting integrables systems
2) Numerical test of the analytic predictions
3) Clear presentation

Weaknesses

No particular weakness

Report

In this manuscript the authors provide an exact formula for the rate function associated with the currents of conserved quantities of a generic non-equilibrium stationary state in integrable systems. Furthermore, they test its validity against Monte Carlo simulations in the classical hard rod gas, and verify that the final result satisfies expected requirements in the case of the interacting Lieb-Liniger model.

The paper is very well written, and the discussions can be easily followed. The results look solid, and are certainly interesting and timely, so that I recommend publication, after the authors have addressed my very minor comments on some aspects of the draft.

As a first comment, even if it is a matter of terminology, I find it a bit misleading to identify the full counting statistics with the scaled-cumulant generating function. Indeed, the latter only gives us access to the large deviation function, namely the fluctuations of the probability around the average, while the full counting statistics provides the exact probability distribution of the current.

Second, in agreement with the first referee, I would soften a few sentences where it is claimed that the results obtained are rigorous. Indeed, I would say that, rather, they are expected to be exact, in the same way as generalized hydrodynamic is.

Finally, in the introduction, and later in the text, the authors state their findings represent the only example where an exact result for a scaled generating function has been obtained for interacting integrable models. While this appears to be true for the case of the currents, other examples are known for different quantities, such as the energy produced after a quench (as done in
[B. Pozsgay, J. Stat. Mech. P10028] for the XXZ Heisenberg chain, or in [arXiv:1904.06259] for the Lieb-Liniger model). Furthermore, an exact formula for the full counting statistics of local densities (for small intervals) and arbitrary stationary states (including the NESSs studied by the authors) was also derived in the Lieb-Liniger model in [Bastianello et al. PRL 120, 190601 (2018), J. Stat. Mech. 113104 (2018)]

---

## Round 3 · Referee Report · Anonymous · 2019-8-2

Strengths

1-New results on the statistics of current fluctuations in interacting integrable systems.
2-Solid numerical verifications.
3-Accessible presentation.

Weaknesses

1-Partial overlap with a closely related paper.

Report

Understanding how to extract fluctuations or the full counting statistics in various interacting quantum dynamical systems is an important problem of statistical mechanics. This work provides some key advancements on this subject. The authors obtain an exact expression for the scaled cumulant generating function for classical and quantum interacting integrable systems, focusing on temporal fluctuations of the total current associated to a local conservation law in an arbitrary homogeneous steady state.

The results of this work concern only the ballistic part, namely fluctuations of currents on Euler time-scale x/t = const. This is achieved via modifying the equilibrium measure with a source term involving the time-integrated current and identifying the flow equation for the Lagrange multipliers as functions of the counting field $\lambda$. The computation requires the signature of the flux Jacobian matrix which can accessed analytically using the generalized hydrodynamics for integrable systems. The $\lambda$-modified state functions can be then computed using the TBA techniques, enabling explicit computation of the scaled cumulant generating function $F(\lambda)$.

The main result of this paper is a closed-form expression for $F(\lambda)$ stated as Eq.(25). The formula is then applied to the interacting classical gas of hard rods (which shows good agreement with Monte Carlo simulations) and the Lieb-Liniger model where the authors numerically verify the fluctuation-dissipation relation for $F(\lambda)$. In addition, the authors derived explicit expressions for the third and fourth cumulant written in terms of hydrodynamic state functions.

I think the paper is well written in general. Unfortunately quite often throughout the paper the authors refer the reader to another closely related paper [88] (by two of the authors) which somewhat disrupts the presentation flow. Apart from that, the paper contains a few original results and therefore definitely warrants publication.

The other referees expressed some concerns regarding the sloppy use of "exact", "rigorous" and "conjectural", and I just want to add my two cents. While I understand the general sentiments, I think one has to simply acknowledge that not all non-rigorous statements are equally conjectural. The formula for currents (17) is simply too transparent and well-established that labelling it as conjectural would be misleading. For instance, the study of spin transport in the XXZ chain where the spin Drude weight derived on the basis of this formula exactly matches the result of the (fully rigorous) operatorial averaging constitutes a non-trivial check outside of relativistic models.

I do not have any major remarks. I hope that the results from Appendix C can be clarified a bit better. There, the authors show that $\lambda$-derivatives at $\lambda=0$ can be substituted by $\mu$-derivative at $\mu = 0$, where here $\mu$ is a small imbalance of the chemical potential drop at the interface. If my understanding holds, this property is implied by the fluctuation-dissipation relation for $F(\lambda)$. In this case I wonder if the implication applies in the opposite direction as well, which would enable to prove (8) without having to invoke (25)?

Typo: I suppose that in eqs 19 and 21 "dr" must be superscript?

Requested changes

- Please clarify the results from Appendix C

---

## Round 4 · Author Response

We would like to thank all the referees as they have clearly made an appreciable effort to engage with our submission. In this message we respond to all the requests and concerns laid out by the reports.

We address the concerns in order, starting with Referee 1. This referee raises two primary issues, the use of the word "exact" and the fact that results deal with homogeneous states despite being generated by the partitioning protocol, a non-homogeneous state.

Concerning the first issue, we would like to point out that the word "exact" is not to be confused with "rigorous". Exact results are results that are supposed to give the exact answer, not an approximation. For this reason, the phrase "exact results" is used widely in the context of integrability, as the techniques give results that are supposed to be exact, not approximate. This is despite the fact that almost all exact results in quantum integrability are *not* rigorous, even if many follow from near-rigorous derivations. Indeed, there are almost no results that we describe as "exact" in the present paper that are actually rigorous (except for some results in stochastic processes). This includes our new results, where full mathematical rigour is lacking. Nevertheless, the results are not approximate, they are exact. Further, much like in integrability, they are not "conjectures" written simply from some physical intuition, they are derived within relatively strong frameworks, that of GHD (now well established) and the ballistic large-deviation theory (whose abstract construction in [92] is as near to mathematically rigorous as many results in quantum integrability). Hence, we would prefer to keep the use of the word "exact". However, in view of the referee's comments we now emphasise, in the introduction, the abstract and the conclusion, that the results are "proposed" or "expected to be exact", but are not rigorously derived. As this is for a physics journal, we hope that it will be clear to the reader that, without the explicit mention of mathematical rigour, the exact results are derived in a non-mathematically-rigorous fashion (as is usual, for instance, in the physics literature on integrability).

The second concern, about the homogeneity of the system, is indeed a very important point that was not clear enough in the previous version. Besides modifications in section 2, page 5 as requested in the list below, we have also made modifications in the introduction in order to clarify the situation. We have added explanations in the paragraph starting with "The states considered are very general, and include the homogeneous current-carrying NESSs obtained by the partitioning protocol..." in order to emphasise that the results apply to homogeneous states generated from inhomogeneous initial conditions. We have also explained the precise check that is done in the hard rod case, where the total charge transfer is evaluated from the initial time of the partitioning protocol (thus, really, in the inhomogeneous situation) -- since we are looking at large-time scaled quantities, the contributions to charge transfer in the emerging large-time homogeneous state dominate. This is mentioned on page 4 (top).

The referee goes on to provide a list of specific fixes which we now turn to in order:

Point 1 is addressed as above; we also changed "theoretical solution" to "hydrodynamic-scale solution". The full Quantum Newton's Cradle (QNC) problem -- with all the experimental effects -- is an open problem, but reproducing from a theoretical framework its main features is no longer an open problem.
Point 2 is addressed above. The addition of "Exact results for transport SCGFs have been obtained in various systems (at various levels of mathematical rigour)", in the introduction, emphasises the point.
Point 3 raises concern about homogeneous GGEs and NESSs. Note that NESSs, despite carrying currents, are stationary and homogeneous -- there is no immediate contradiction. But indeed the situation was not made clear enough. We have added more explanations in section 2, page 5, especially stating how a homogeneous state emerges from the inhomogeneous initial condition.
Point 4 is related to the use of the word exact in constructing NESSs from the partitioning protocol, and is answered above; we took away the word "exactly" in section 2, page 5.
Point 5 -- Perhaps this was the cause of all the problems about language -- this was a typo! It is certainly the case that our results are not rigorous. We apologise for the omission of the word "not" in the sentence identified by the referee, this has been rectified.

Referee 2 also raises issues regarding the use of "exact" which we have addressed through the corrections made above. The second issue raised was that of identifying full counting statistics with the scaled-cumulant generating functional (SCGF). In the literature, the terminology "full counting statistics" has been used since the results of Levitov and Lesovik to represent the *scaled* cumulant generating function in transport -- the Levitov-Lesovik formula is an SCGF, as are the formulae of Saleur in integrable impurity models. Hence our nomenclature is consistent with this long-term convention; the use of "full counting statistics" in recent works on other types of counting problems deviates from this established convention. For this reason, we would prefer to keep the terminology. A footnote has been added to this effect on page 3. Finally the referee notes that other results exist for interacting integrable models. We would like to emphasise that our claim is that we have obtained the first transport SCGFs; indeed there are other SCGFs that can be, and have been, calculated, but they are not transport characteristics. We have further added "model-agnostic" in the introduction to emphasise that, contrary to many previous results, the formula we have obtained holds for a vast range of interacting models, including the models mentioned by the referee. Notwithstanding this, we have added a sentence to refer to other, non-transport counting statistics calculated in particular models, with the proposed references.

Referee 3 explains that our previous use of "exact" is not incorrect, but states they understand the general sentiment of the other referees. Thus the changes above will also address this referee's comments.

Concerning the reliance on [92] (previously [88]), we would like to quickly point out that the general theory's derivation is relatively involved; while, at the same time, the application to integrability is nontrivial (requiring the full structure of GHD), and it is probably one of the most interesting applications; thus the separation.

The major concern raised here is regarding the formulation of Appendix C. We have re-written Appendix C, hopefully making it clearer. We also would like to thank the referee for comments on proving fluctuation relations using the linear response theory presented in Appendix C. We had thought of this approach but a solution has eluded us thus far. The problem is that this linear response formulation still does not provide a solution for finite $\lambda$; it is still an order-by-order construction. Also, even looking for the order-by-order fluctuation relations (i.e. expanding in $\mu$ and $\lambda$), we would need to have relations involving higher-order $\mu$ derivatives, while what we have established only involves the first order expansion in $\mu$. This direction might eventually be fruitful, but a bit more work is necessary.

Finally there was concern of a typo in equations (19) and (21) - this was a notational issue and an explanation has been added for clarification in a footnote on page 10.

We hope that this response sufficiently addresses the concerns of the referees. Once again we would like to thank the referees for producing such relevant comments which must have required a very thorough reading of this submission. We also would like to thank the editor for considering these responses.

Regards,
Jason, Joe, Rosemary and Benjamin.

---

## Round 4 · List of Changes

- Adjusted use of the qualifier "exact" and made distinction with "rigorous" in the introduction and conclusion.

- Clarified the need for homogeneity and the specific setup we test the theory on in the introduction. Clarified the meaning of NESS, and the specific of NESS our theory applies to

- Added references as aked by the referees

- Re-wrote appendix C in order to clarify it

You are currently on this page

Resubmission 1812.02082v4 on 26 November 2019

---

## Editorial Decision

published